# Observation of topological superconductivity in a stoichiometric transition metal dichalcogenide 2M-WS$_2$

Y. W. Li[1,2,3,21], H. J. Zheng[1,3,4,21], Y. Q. Fang[5,6,21], D. Q. Zhang[7,8,9,21], Y. J. Chen[10], C. Chen[1,3,11], A. J. Liang[1,3], W. J. Shi [12,13], D. Pei[2], L. X. Xu[1,4], S. Liu[1,4], J. Pan[5], D. H. Lu [14], M. Hashimoto [14], A. Barinov [15], S. W. Jung [16,17], C. Cacho [16], M. X. Wang[1,3], Y. He[18], L. Fu[19], H. J. Zhang [8,9✉], F. Q. Huang [5,6✉], L. X. Yang[10,20✉], Z. K. Liu [1,3✉] & Y. L. Chen[1,2,3,10✉]

Topological superconductors (TSCs) are unconventional superconductors with bulk super-conducting gap and in-gap Majorana states on the boundary that may be used as topological qubits for quantum computation. Despite their importance in both fundamental research and applications, natural TSCs are very rare. Here, combining state of the art synchrotron and laser-based angle-resolved photoemission spectroscopy, we investigated a stoichiometric transition metal dichalcogenide (TMD), 2M-WS$_2$ with a superconducting transition temperature of 8.8 K (the highest among all TMDs in the natural form up to date) and observed distinctive topological surface states (TSSs). Furthermore, in the superconducting state, we found that the TSSs acquired a nodeless superconducting gap with similar magnitude as that of the bulk states. These discoveries not only evidence 2M-WS$_2$ as an intrinsic TSC without the need of sensitive composition tuning or sophisticated heterostructures fabrication, but also provide an ideal platform for device applications thanks to its van der Waals layered structure.

[1] School of Physical Science and Technology, ShanghaiTech University, Shanghai 201210, People's Republic of China. [2] Department of Physics, University of Oxford, Oxford OX1 3PU, UK. [3] ShanghaiTech Laboratory for Topological Physics, Shanghai 201210, People's Republic of China. [4] University of Chinese Academy of Sciences, Beijing 100049, People's Republic of China. [5] State Key Laboratory of High Performance Ceramics and Superfine Microstructure Shanghai Institute of Ceramics Chinese Academy of Science, Shanghai 200050, People's Republic of China. [6] State Key Laboratory of Rare Earth Materials Chemistry and Applications College of Chemistry and Molecular Engineering, Peking University, Beijing 100871, People's Republic of China. [7] School of Physics, China Jiliang University, Hangzhou 310018, People's Republic of China. [8] National Laboratory of Solid State Microstructures and School of Physics Nanjing University, Nanjing 210093, People's Republic of China. [9] Collaborative Innovation Center of Advanced Microstructures, Nanjing 210093, People's Republic of China. [10] State Key Laboratory of Low Dimensional Quantum Physics, Department of Physics, Tsinghua University, Beijing 100084, People's Republic of China. [11] Advanced Light Source, Lawrence Berkeley National Laboratory, Berkeley, CA 94720, USA. [12] Center for Transformative Science, ShanghaiTech University, Shanghai 201210, People's Republic of China. [13] Shanghai high repetition rate XFEL and extreme light facility (SHINE), ShanghaiTech University, Shanghai 201210, People's Republic of China. [14] Stanford Synchrotron Radiation Lightsource, SLAC National Accelerator Laboratory, Menlo Park, CA 94025, USA. [15] Elettra-Sincrotrone Trieste, Trieste, Basovizza 34149, Italy. [16] Diamond Light Source, Harwell Campus, Didcot OX11 0DE, UK. [17] Department of Physics, Gyeongsang National University, Jinju 52828, Korea. [18] Department of Physics, University of California at Berkeley, Berkeley, CA 94720, USA. [19] Department of Physics, Massachusetts Institute of Technology, Cambridge, MA 02139, USA. [20] Frontier Science Center for Quantum Information, Beijing 100084, People's Republic of China. [21] These authors contributed equally: Y. W. Li, H. J. Zheng, Y. Q. Fang, D. Q. Zhang. ✉email: zhanghj@nju.edu.cn; huangfq@mail.sic.ac.cn; lxyang@tsinghua.edu.cn; liuzhk@shanghaitech.edu.cn; yulin.chen@physics.ox.ac.uk

Superconductors and topological matter are two classes of quantum materials with fascinating physical properties[1,2] and application potentials[3,4], and their combination may further give rise to a unique quantum phase, the topological superconductor (TSC). A TSC can host exotic emergent particles such as the Majorana fermion[1,2], a particle of its own anti-particle that can not only show remarkable phenomena such as thermal quantum Hall effect[5], but also serve as a key ingredient for the realization of topological quantum computation—a promising approach to realize the fault-tolerant quantum computation[1–4,6].

However, TSCs are rare in nature. Up to date, intrinsic TSC candidates are only found in a few materials, which are either controversial (e.g., $Sr_2RuO_4$[7,8]) or nonstoichiometric compounds (e.g., $Cu_xBi_2Se_3$[9,10], $FeTe_xSe_{1-x}$[11–13], and $Li(Fe,Co)As$[14]) that require fine tuning in composition and inevitably consist of defects[13,15] unfavorable for device applications. On the other hand, artificial structures combining conventional superconductors and nondegenerate spin states (e.g., topological insulators[16–19], quantum anomalous hall insulators[20], semiconductors with strong spin–orbit coupling[21–23], or ferromagnetic thin film and atomic chains[24,25]) have been explored, but the need to construct sophisticated heterostructures and the requirement of long superconducting coherence length make this approach material selective.

Recently, a stoichiometric transition metal dichalcogenide (TMD), 2M-WS$_2$ was proposed as an intrinsic TSC with the highest intrinsic superconducting transition temperature ($T_C$ =

8.8 K) among all TMDs in the natural form[26,27]. The stoichiometry makes the synthesis of high-quality crystals possible, and the layered structure with van der Waals coupling makes it ideal for device fabrication for potential applications.

The superconductivity of 2M-WS$_2$ was recently confirmed by transport[26] and scanning tunneling microscopy/spectroscopy (STM/STS) measurements[27], and zero energy peaks in the STS spectra were observed in magnetic vortex cores[27], suggesting the possible existence of Majorana bound states (MBSs). However, compelling evidence for the topological electronic structure, including the direct observation of topological surface states (TSSs), the superconducting gap of the TSSs and its temperature evolution, are yet to be found.

In this work, combining the state of the art synchrotron- and laser-based angle-resolved photoemission spectroscopy (ARPES), we not only systematically investigated the band structure of 2M-WS$_2$ across the full three-dimensional Brillouin zone (BZ) thanks to the large photon energy range of the synchrotron light source, but also successfully observed the TSSs thanks to the high energy and momentum resolution made possible by the laser light source. Furthermore, by carrying out detailed temperature-dependent measurements, the superconducting gap from the TSSs (as well as the bulk states, BSs) was clearly observed below $T_C$, establishing the TSC nature of 2M-WS$_2$.

The mechanism of the topological superconductivity in 2M-WS$_2$ can be briefly shown in Fig. 1a. Above the superconducting transition ($T > T_C$), 2M-WS$_2$ is a bulk topological semimetal

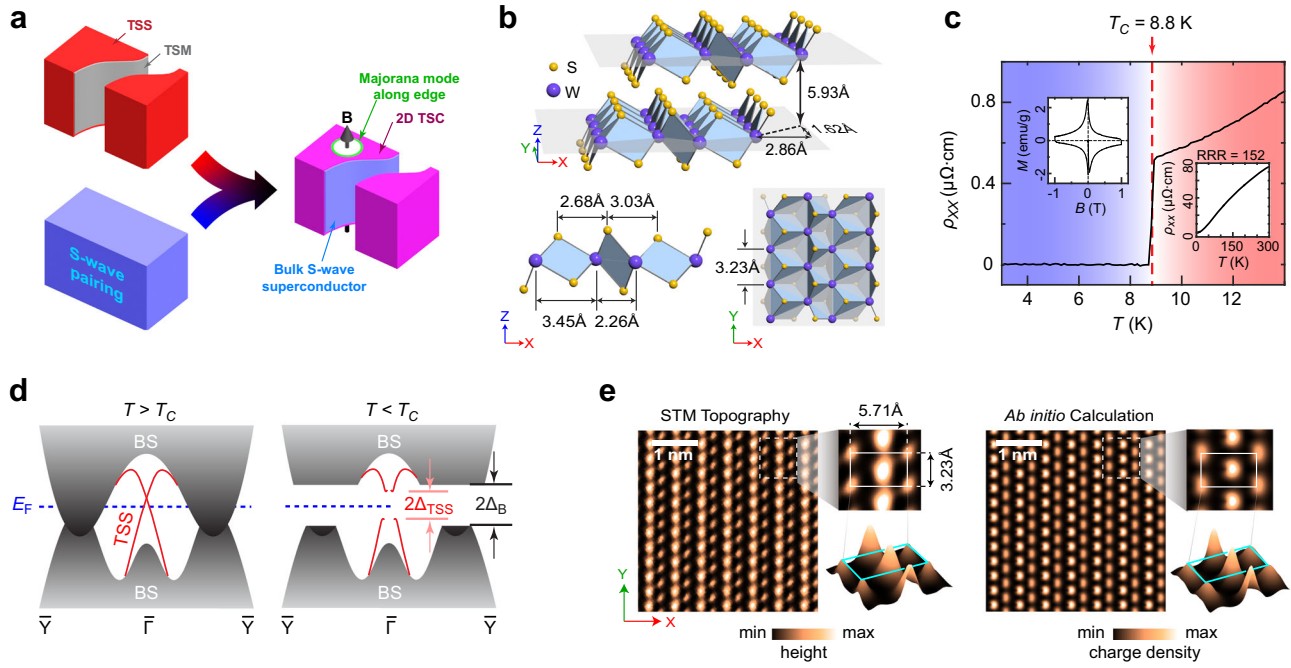

**Fig. 1 Introduction to topological superconductivity and general characterizations of 2M-WS$_2$. a** Illustration on the formation of topological superconductivity in 2M-WS$_2$ (see text for details). Top panel: Topological surface state (TSS, red color) and the bulk topological semimetal (TSM, gray color) of 2M-WS$_2$. Bottom left panel: Bulk s-wave superconductivity at $T < T_C$. Right panel: Surface (2D) topological superconductivity (TSC, magenta color) of TSS, induced by the internal proximity effect from the bulk s-wave superconductivity. Majorana state (green circle) can exist at the magnetic vortex core (illustrated by the black arrow and white color region). **b** Top panel: 3D view of the crystal structure, showing two adjacent WS$_2$ layers offset in the X- and Y-directions. Bottom left panel: Side view. Bottom right panel: Top view of a WS$_2$ layer. **c** Temperature dependence of resistance shows a sharp superconducting transition at $T_C = 8.8$ K. Left inset: Hysteresis loop of magnetization curve at 2 K showing the Meissner effect and the threshold field value. Right inset: Resistance curve at a large temperature range (3-300 K), showing a residual-resistivity ratio (RRR) of 152. **d** Illustration of the band structure of 2M-WS$_2$ in the normal (left) and superconducting states (right). The bulk states (BSs) is shown in solid gray color and the TSSs as solid red lines. In the superconducting states, the BSs and TSSs are both gapped (more details can be found in Fig. 4 below). $E_F$, Fermi energy. $\Delta_{TSS}$ and $\Delta_B$, superconducting gap of the TSS and the BS, respectively. **e** Left panel: Surface topography of a cleaved sample from scanning tunneling microscopy (STM) measurement shows the perfect topmost S-atom layer without defects. Zoomed in plots with more information can be found in insets on the right. Right panel: Ab initio calculation of the charge density from the topmost S-atom layer shows excellent agreement with the STM measurement.

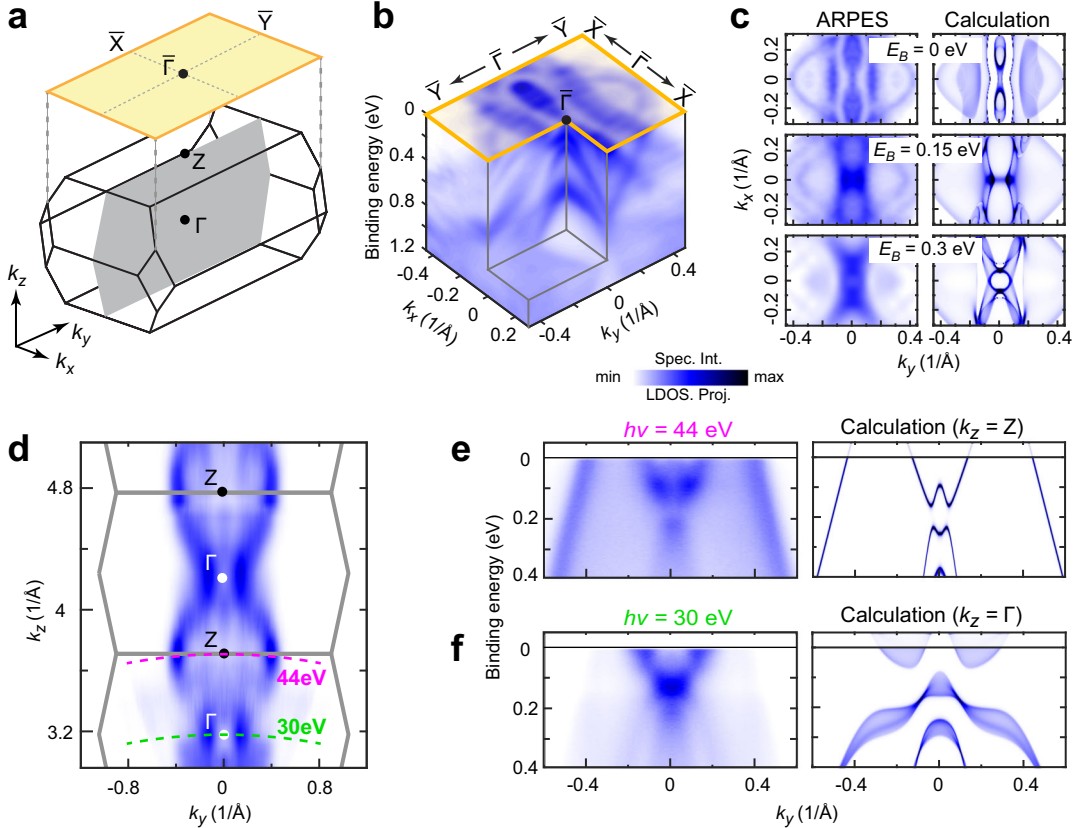

**Fig. 2 Overall electronic structures across the 3D Brillouin zone. a** Schematic of the bulk (bottom) and (100) surface (top) Brillouin zone (BZ) of 2M-WS$_2$ with high symmetry points labeled. The gray plane in the bulk BZ indicates the $k_y$–$k_z$ measurement plane in **d**. **b** 3D intensity plot of photoemission spectra around the $\bar{\Gamma}$ point, showing the band dispersions and the resulting FSs. **c** Comparison showing nice agreement between experiments (left) and ab initio calculations (right) of three constant energy contours at different binding energies. For better comparison with calculations, the experimental plot has been symmetrized with respect to the $k_y = 0$ plane according to the crystal symmetry (same below). **d** Photoemission intensity plot of the $k_y$–$k_z$ plane ($k_x = 0$, the gray plane in **a**), showing clear $k_z$ dispersion (and thus the 3D nature) of 2M-WS$_2$. Green and magenta curves indicate the $k_z$ momentum loci (probed by 30 eV and 44 eV photons, respectively) of the two band dispersions shown in (**e**) and (**f**), respectively. **e**, **f** Comparisons between photoemission dispersions (left) and corresponding calculations (right) cutting across high symmetry $k_z$ loci (around $\bar{\Gamma}$ and $Z$ points, respectively, and $k_z$ Integration window around $\Gamma$ and $Z$ points is 0.06 $\pi/a$, where $a$ is the out-of-plane lattice constant) in the bulk BZ as indicated in **d**.

(TSM) with nontrivial TSSs; while in the superconducting state ($T < T_C$), the bulk s-wave superconductivity can induce superconductivity to the TSSs by internal proximity effect, thus realizing a topologically nontrivial superconducting state of spin-helical electrons on the surface, as illustrated in Fig. 1a. In addition, if a magnetic domain is formed, the domain boundary (green line in the right panel of Fig. 1a) can host the MBS[1,2,6], which, if bound with a magnetic flux quantum, obey non-abelian statistics[1,2] and can be used to form the quantum-bit for topological quantum computation[1–4,6]. Similar self-proximity-induced TSC states have been proposed in iron-based superconductors[11–14] and PbTaSe$_2$[28].

## Results

**Sample characterization.** Like many TMD compounds, 2M-WS$_2$ is a van der Waals layered crystal (space group $C_{2/m}$, No. 12). As shown in Fig. 1b, within each WS$_2$ layer, the distorted octahedral structure forms quasi-1D chains of W and S atoms along $Y$ direction (see Fig. 1b); and the adjacent WS$_2$ layers are offset to each other along $X$- and $Y$-directions by $\Delta X = 2.86$ Å and $\Delta Y = 1.62$ Å, respectively (Fig. 1b, and more details on the crystal structure are shown in Supplementary Fig. 1). The high quality of the crystals used in this work was confirmed by X-ray diffraction (Supplementary Fig. 2), the large residual-resistivity ratio (RRR = 152, Fig. 1c, right inset), and the sharp superconducting transition

($\Delta T = 0.2$ K, Fig. 1c). In fact, the $T_C$ (8.8 K) of 2M-WS$_2$ is the highest among all intrinsic TMD compounds under ambient conditions up to date[26].

The nontrivial topological electronic structure of 2M-WS$_2$ was theoretically proposed in a recent work[26,29] and is confirmed by our ab initio calculations: 2M-WS$_2$ is a TSM in its normal state with the TSSs located at the center of the surface BZ (sketched in the left panel of Fig. 1d); in the superconducting state (right panel of Fig. 1d), both the BSs and TSSs are expected to be gapped (as observed in our measurements which will be discussed in details later).

Due to the weak van der Waals interaction between WS$_2$ layers, the cleaved surfaces for the ARPES study in this work are of high quality. As illustrated in Fig. 1e, a topography map from our STM measurement (left panel of Fig. 1e) reveals a defect-free region of the topmost S-atom layer with clear quasi-1D chains, in excellent agreement with the ab initio simulation (right panel of Fig. 1e and more details in Supplementary Fig. 3).

**General electronic structure of 2M-WS$_2$.** To investigate the overall electronic structure of 2M-WS$_2$, we first carried out synchrotron-based ARPES measurements, and Fig. 2b shows a typical set of experimental band structure, containing rich details that can be successfully reproduced by ab initio calculations (see Fig. 2c and Supplementary Fig. 4 for more details). Thanks to the

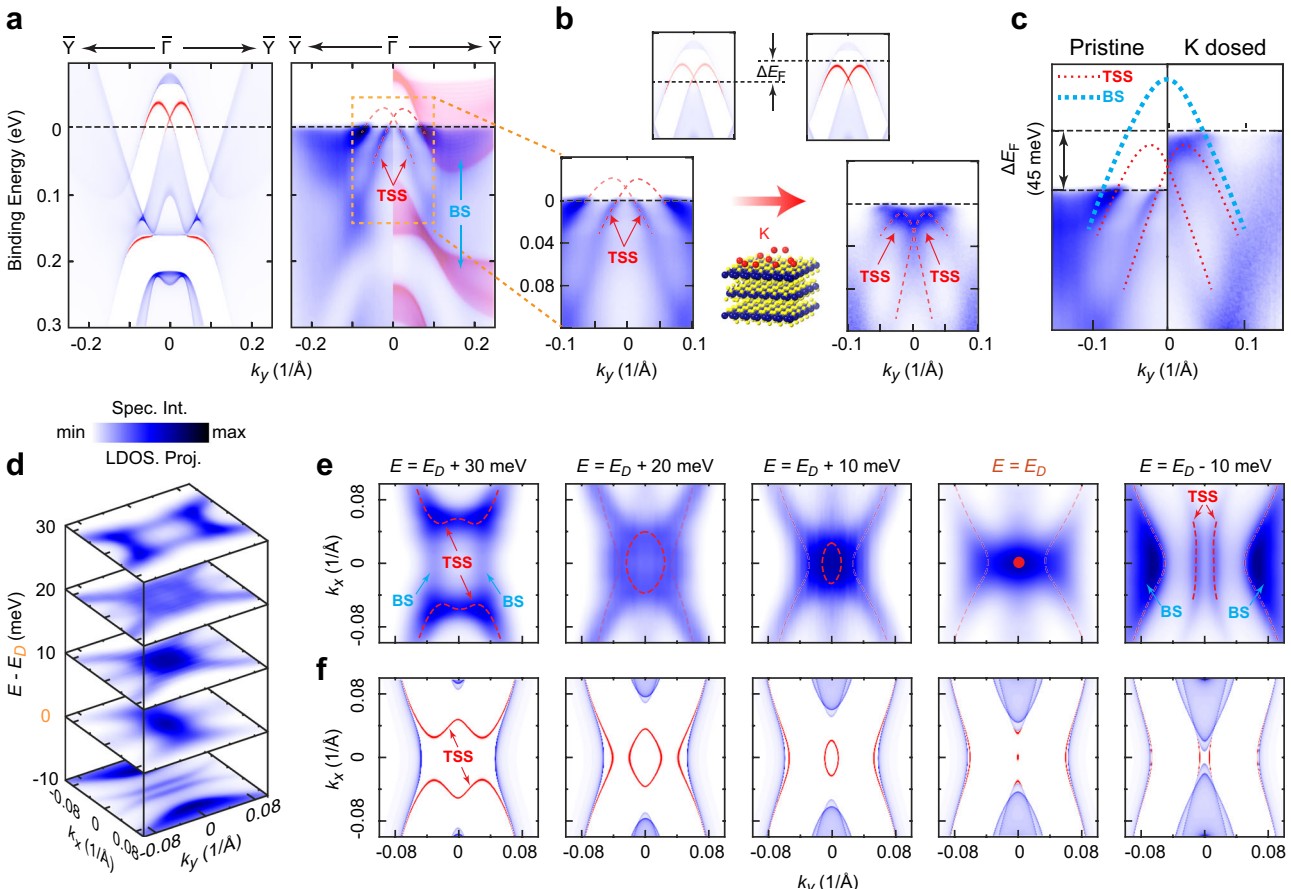

**Fig. 3 Band structure of the topological surface states. a** Comparison between slab calculation (left) and photoemission (right) band dispersions along $\bar{Y} - \bar{\Gamma} - \bar{Y}$ direction. The TSSs are highlighted by red color (left) and indicated by the red dashed lines (right). The bulk dispersions around $\Gamma$ point (red shaded area, $k_z$ integration window is 0.06 $\pi/a$) are overlapped to better illustrate the TSSs (right). **b** Bottom left: Zoomed in dispersion of the TSS in the pristine sample covering the yellow box area in the right panel of **a**. The red dashed lines show the TSS dispersions. Top left: Schematic of the calculated dispersions for comparison and the $E_F$ is marked as a black dashed line. Bottom right: Full dispersion of the TSS band observed after in situ potassium (K) doping (illustrated as the middle inset). Top right: Calculated dispersions with the new $E_F$ position indicated after the K-dosing, showing a clear upshift $\Delta E_F$. **c** Side-by-side comparison of TSS dispersions before (left panel) and after (right panel) K-dosing, shows a $\Delta E_F$ of ~45 meV. **d** Stacking plots of the experimental constant energy contours (CECs) at different energy close to the Dirac point, showing rapid evolution. **e, f** Comparison between experimental (**e**) and ab initio calculated (**f**) CECs evolution near the Dirac point showing overall agreement. For clarity, the CECs of the TSSs are highlighted by red color. In **e**, the faded red dashed lines represent the less visible TSSs due to the hybridization to the bulk states.

wide photon energy range of the synchrotron light source, we were able to do detailed photon-energy-dependent measurements to examine the electronic structures across multiple 3D BZs along the $k_z$ direction, as illustrated in Fig. 2d (see Supplementary Fig. 5). With the $k_z$ electronic structures identified, we can study band dispersions at different $k_z$ momenta using specific photon energies. As two examples, the dispersions cut across the bulk $\Gamma$ ($k_z = 0$) and $Z$ ($k_z = \pi/a$, where $a$ is the out-of-plane lattice constant) points can be accessed by 30 and 44 eV photons, respectively (see Fig. 2d); and the comparison between experiments and ab initio calculations (Fig. 2e, f) shows nice overall consistency (see Supplementary Fig. 6).

**Topological surface states of 2M-WS₂.** According to our calculation and the previous theory work[26], the TSSs in 2M-WS₂ reside only in a close vicinity around the center ($\bar{\Gamma}$ point) of the surface BZ, thus very high energy and momentum resolution is needed to resolve the TSS band structures. Therefore, we adopted the laser-based ARPES ($h\nu = 6.994$ eV, $\Delta E = 0.9$ meV and $\Delta k = 0.003$ Å$^{-1}$) in the search for TSSs.

Indeed, as presented in Fig. 3a, the TSSs near the $\bar{\Gamma}$ in the calculation (left panel) were clearly seen in measurement (right panel). However, in the pristine sample, the upper part of the TSSs was above the Fermi-level ($E_F$) and could not be seen experimentally, and slight n-type doping was needed to lift the $E_F$ to reveal the whole TSS dispersions. As expected, after in situ potassium doping, we successfully raised the $E_F$ by ~45 meV and the whole TSS dispersions could be observed (Fig. 3b, c), in excellent agreement with the calculation. Besides the dispersions, the existence of the TSSs can also be seen in the constant energy contours of the band structures (Fig. 3d–f), again in good agreement with the calculations.

**Superconducting gap of 2M-WS₂.** Having identified the TSSs, we then investigate the superconducting state properties of the TSSs and BSs. The superconducting gaps of both the TSSs and BSs from a pristine sample are clearly seen in the left panel of Fig. 4a (spectra were symmetrized with respect to $E_F$ to get rid of the Fermi-Dirac function cutoff), which disappear when $T > T_C$ (right panel of Fig. 4a). The evolution of the superconducting gap

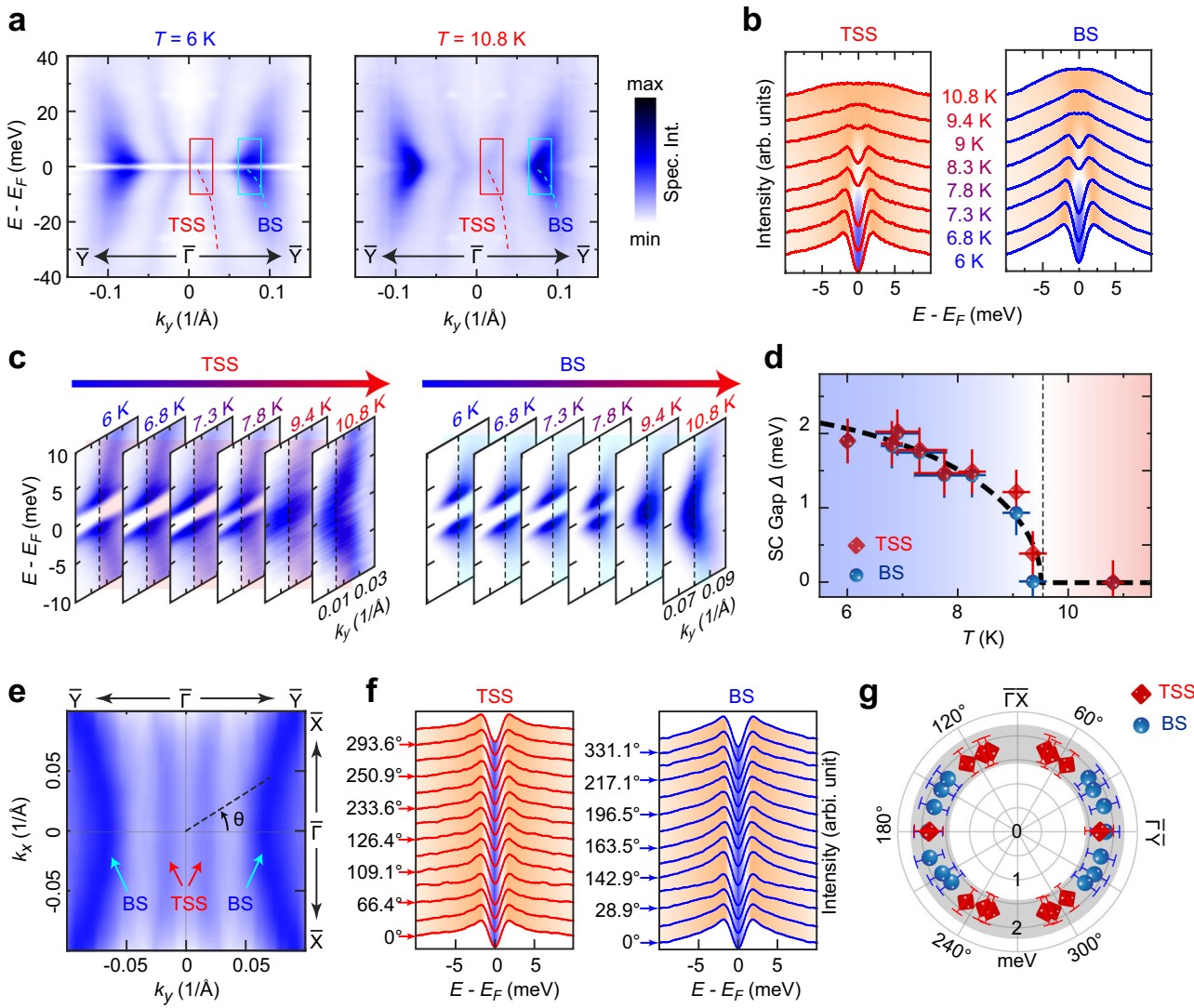

**Fig. 4 Superconducting gap from the TSS and BS. a** Photoemission spectra intensity plots of the band dispersions along $\bar{Y} - \bar{\Gamma} - \bar{Y}$ direction in the superconducting (left) and normal (right) states show clear superconducting gaps from both the TSS (red dashed lines) and BS (blue dashed lines, which show clear $k_z$ broadening and form the bulk $\alpha$ pocket in Supplementary Fig. 5). Note that to better illustrate the gap, the spectra are symmetrized in energy direction with respect to $E_F$ (same in plots **b, c, f** below). **b** Temperature dependence of the energy distribution curves (EDCs) at the Fermi momentum ($\mathbf{k}_F$) of TSS (left) and BS (right) shows the opening of the superconducting gap below $T = 9$ K. **c** Temperature dependence of the band dispersions of the TSS (left) and BS (right) across $\mathbf{k}_F$ (the momentum and energy range plotted here corresponds to the red and cyan rectangle area marked in **a**, again shows the clear superconducting gap below $T_C$. **d** Temperature evolution of the superconducting gap from the TSS and BS (details on the gap extraction from the AREPS spectra function can be found in Supplementary Fig. 7) is consistent with a mean field temperature dependence, as indicated by the black dashed line. Horizontal error bars represent the temperature variance during the measurements. Vertical error bars represent the confidence intervals of the fitting under 95% confidence level. **e** Constant energy contour plot (intensity integrated between $E_F$ to $E_b = 10$ meV) shows the $\mathbf{k}_F$ loci of the TSS and BS ($T = 6$ K). $\theta$ defined the angle used in **f, g**. **f** EDCs from the $\mathbf{k}_F$ loci of different $\theta$ angles at the superconducting state ($T = 6$ K) show that the superconducting gap magnitude of both TSS (left) and BS (right) are nearly isotropic. **g** Plot of the superconducting gap size as a function of $\theta$ angle for both TSS (red symbols) and BS (blue symbols), showing similar magnitude. Error bars represent the energy resolution determined by ARPES spectra on multi-crystalline gold.

with temperature can be seen in the energy distribution curves (EDCs, see Fig. 4b) or directly from the spectra plot (Fig. 4c). The superconducting gaps from both the TSSs and BSs are consistent with a mean field temperature dependence (Fig. 4d, more details about the gap extraction can be found in Supplementary Fig. 7).

Finally, benefited from the inherent momentum resolution of ARPES measurements, we were able to study the in-plane superconducting gap distribution (Fig. 4e–g), which shows a nodeless s-wave-like symmetry for both TSSs and BSs, making 2M-WS₂ an ideal system of 2D TSC on the surface[16]. In addition, the s-wave gap and the velocity anisotropy near the Dirac point of

TSSs revealed in our measurements (see Fig. 3e and Supplementary Fig. 8) suggest an anisotropy of the MBSs: the angular distribution of $\sim h v_F/\Delta$ (where $h$, $v_F$, and $\Delta$ are the Planck constant, Fermi-velocity, and the superconducting gap, respectively) from our experiment provides a direct momentum-resolved perspective that explains the recent STM report of such anisotropy[27].

## Discussion

Our unambiguous observation of superconducting gap developed on the TSSs of 2M-WS₂ provide the key ingredient for TSC based

on the Fu-Kane proposal[2,16]. The helical spin texture of the TSS is confirmed by previous first principle calculation[26] and should survive possible surface-bulk hybridization[30,31] since it is well separated from the BS in both energy and momentum space. Moreover, spin polarization persists in the TSSs of n-doped topological insulators[30,31], topological metals[32], and even the topological surface resonance states of Dirac semimetals[33], which exhibits robustness of spin texture in TSSs against surface-bulk hybridization. Therefore, the spin-momentum locked surface states could result in effective p-wave superconductivity in 2M-$WS_2$, namely a TSC state.

As shown in Fig. 3a–c, 2M-$WS_2$ hosts a Rashba-like TSS, which is a reminiscence of semiconductor–superconductor TSC scheme. As the upper and lower branches of the TSS must eventually merged in the spin unpolarized BSs, the $E_F$ position is critical: (1) when only one pair of spin-polarized bands crosses $E_F$, 2M-$WS_2$ falls to the superconducting topological insulator scenario, as proposed in $Cu_xBi_2Se_3$;[9,10] whereas (2) two pairs of spin-polarized bands cross $E_F$, 2M-$WS_2$ falls to the strong spin–orbit coupling semiconductor–superconductor scenario, as proposed in Al-InAs heterostructure[23]. For the latter situation, the topology of the system can be tuned by the Zeeman coupling $V_Z$, which is due to the external magnetic field as shown in Fig. 1a, chemical potential $\mu$, and the self-proximity-induced pairing gap $\Delta_0$[21,34]. The compatibility and versatility of the TSS configuration in 2M-$WS_2$ with the appearance of Majorana states require further investigation.

The formation of the TSC state in stoichiometric 2M-$WS_2$, together with its layered structure with van der Waals coupling, makes it ideal for device fabrication and thus a promising platform to explore the phenomena of MBSs and their application in topological quantum computation. Moreover, the discovery of the TSC state in 2M-$WS_2$ further enriches the interesting phenomena hosted by TMDs (e.g., gate and pressure-tunable superconductivity[19,35–39], charge density waves[38–41], Mott insulators[38,39], gyrotropic electronic orders[42], moiré-trapped valley excitons[43], type-II Dirac and Weyl semimetals[44–46], etc.), thus enabling the study of these properties and their interplay, as well as the design of novel devices for new applications. We note that, after the acceptance of this manuscript, self-proximity induced TSC was also theoretically proposed in $RRuB\_2$ (R=Y, Lu)[47].

## Methods

**Sample preparation**. 2M-$WS_2$ single crystals were prepared by the deintercalation of interlayer potassium cations from $K_{0.7}WS_2$ crystals. For the synthesis of $K_{0.7}WS_2$, $K_2S_2$ (prepared via liquid ammonia), W (99.9%, Alfa Aesar) and S (99.9%, Alfa Aesar) were mixed by the stoichiometric ratios and ground in an argon-filled glovebox. The mixtures were pressed into a pellet and sealed in the evacuated quartz tube. The tube was heated at 850 °C for 2000 min and slowly cooled to 550 °C at a rate of 0.1 °C/min. The synthesized $K_{0.7}WS_2$ (0.1 g) was oxidized chemically by $K_2Cr_2O_7$ (0.01 mol/L) in aqueous $H_2SO_4$ (50 mL, 0.02 mol/L) at room temperature for 1 h. Finally, the 2M-$WS_2$ crystals were obtained after washing in distilled water for several times and drying in the vacuum oven at room temperature[26].

**Angle-resolved photoemission spectroscopy**. Synchrotron-based ARPES data were taken at Spectromicroscopy of Elettra Synchrotron, Italy (proposal no. 20190294), beamline I05 of Diamond Light Source (DLS), UK (proposal no. SI125135-1), and beamline BL5-2 of Stanford Synchrotron Radiation Laboratory (SSRL), Stanford Linear Accelerator Center (SLAC), USA (proposal no. 5069). The samples were cleaved in situ and aligned the $\bar{\Gamma} - \bar{Y}$ direction parallel to the analyzer slit. The measurements were under ultra-high vacuum below $5 \times 10^{-11}$ Torr. Data were collected by an internal movable electron energy analyzer at Spectromicroscopy of Elettra Synchrotron, a Scienta R4000 analyzer at I05 beamline of DLS, UK, and a DA30L analyzer at beamline BL5-2 of SSRL, SLAC, USA. The total energy resolutions were 30 meV at Spectromicroscopy, and below 10 meV at beamline I05 of DLS and beamline BL5-2 of SSRL, respectively. The angle resolution was 0.2°.

High-resolution laser-based ARPES measurements were performed at home-built setups ($h\nu = 6.994$ eV) at ShanghaiTech University and Tsinghua University. The samples were cleaved in situ and aligned the $\bar{\Gamma} - \bar{Y}$ direction parallel to the

analyzer slit. The measurements were under ultra-high vacuum below $5 \times 10^{-11}$ Torr. Data were collected by a DA30L analyzer. The total ultimate energy and angle resolutions were 0.9 meV and 0.2°, respectively.

**Scanning tunneling microscopy/spectroscopy**. In the STM experiment, cleaved single crystals were transferred to a cryogenic stage in the ultra-high vacuum (<$2 \times 10^{-10}$ Torr) and kept at 80 K. Pt-Ir tips were used for imaging, which were all decorated and calibrated on the surface of silver islands grown on p-type Si (111) $-7 \times 7$.

**Single-crystal X-ray diffraction**. Single-crystal XRD was performed using Mo target by the Rigaku Oxford Diffraction at the Department of Physics, University of Oxford. The beam spot size is 10–200 μm in diameter. The data were collected and analyzed by the CrysAlisPro software.

**Transport measurement**. Transport measurements were taken in the Physical Properties Measurement System (PPMS) of Quantum design.

**First principle calculations**. The first principle calculations were carried out in the framework of the generalized gradient approximation (GGA) functional of the density functional theory through employing the Vienna ab initio simulation package (VASP) with the projector augmented wave pseudopotentials[48]. The experimental lattice constants were taken, and inner positions were obtained through full relaxation with a total energy tolerance $10^{-5}$ eV. The SOC effect was self-consistently included. Modified Becke–Johnson (mBJ) functional was employed for the double check and the electronic structure and topological nature remains unchanged.

## Data availability

The datasets that support the findings of this study are available from the corresponding author upon reasonable request. CCDC 1853656 contains the crystallographic data in the Supporting Information for this paper. These data can be obtained free of charge via www.ccdc.cam.ac.uk/data_request/cif, or by emailing data_request@ccdc.cam.ac.uk, or by contacting The Cambridge Crystallographic Data Centre, 12 Union Road, Cambridge CB21EZ, UK; fax: +44 1223 336033.

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

## Acknowledgements

The work is supported by the National Key R&D program of China (Grants No. 2017YFA0305400). Y.L.C. acknowledges the support from the Oxford-ShanghaiTech collaboration project and the Shanghai Municipal Science and Technology Major Project (grant 2018SHZDZX02). Z.K.L. acknowledges the support by Shanghai Technology Innovation Action Plan 2020-Integrated Circuit Technology Support Program (Project No. 20DZ1100605). L.X.Y. acknowledges the support by the Natural Science Foundation of China (Grants No. 11774190 and No. 11634009). H.J.Z. acknowledges the support by the Fundamental Research Funds for the Central Universities (Grant No. 020414380149), Natural Science Foundation of Jiangsu Province (No. BK20200007), the Natural Science Foundation of China (Grants No. 12074181 and No. 11834006) and the Fok Ying-Tong Education Foundation of China (Grant No. 161006). Y.W.L. acknowledges the support from International Postdoctoral Exchange Fellowship Program (Talent-Introduction Program, Grants No. YJ20200126). D.Q.Z. acknowledges the support by the National Natural Science Foundation of China (No. 12004361). Y.Q.F. acknowledges the support by the Natural Science Foundation of China (Grant No. 21871008). D.P acknowledges the support from China Scholarship Council. Use of the Stanford Synchrotron Radiation Lightsource, SLAC National Accelerator Laboratory, is supported by the U.S. Department of Energy, Office of Science, Office of Basic Energy Sciences under Contract No. DE-AC02-76SF00515. Some of the calculations were carried out at the HPC Platform of ShanghaiTech University Library and Information Services and at the School of Physical Science and Technology.

## Author contributions

Y.L.C. conceived the experiments. Y.W.L. and H.J. Zheng carried out ARPES measurements with the assistance of Z.K.L., L.X.Y., Y.J.C., C.Chen, A.J.L., D.P., L.X.X., D.H.L., M.H., A.B., S.W.J., C.Cacho, and Y.H.; Y.W.L. and H.J.Zheng performed the data analysis on the ARPES results. Y.Q.F., J.P., and F.Q.H. synthesized and characterized the bulk single crystals. D.Q.Z., W.J.S., and H.J.Zhang performed ab initio calculations. H.J.Zheng, S.L., and M.X.W. carried out STM measurements. Y.W.L. wrote the first draft of the paper. Y.L.C., Z.K.L., L.X.Y., H.J.Zhang, Y.H., and L.F. contributed to the revision of the manuscript. All authors contributed to the scientific planning and discussions.

## Competing interests

The authors declare no competing interests.
