## [Peer Review File · Nature Communications]

REVIEWERS' COMMENTS

Reviewer #1 (Remarks to the Author):

The authors have addressed my questions in the previous report, and I recommend publication of the current manuscript in Nature Communication.

Reviewer #3 (Remarks to the Author):

The authors have addressed the reviewers' criticism in their detailed rebuttal and in the manuscript. I agree with the other reviewers that this work does not report an absolute first observation nor present a definitive "smoking gun" for topological superconductivity. Nevertheless, it does represent a good step forward owing to the excellent momentum and energy resolution and the careful temperature dependence. The analysis appears to be sound. I believe that the paper will be of interest to the readers of Nature Communications.

Marco Gioni

Reply to the Referees' Comments

Reviewer #1:

The authors have addressed my questions in the previous report, and I recommend publication of the current manuscript in *Nature Communication*

Authors' reply:

We thank the reviewer for the recommendation of publication of the current manuscript in *Nature Communications*.

Reviewer #3:

The authors have addressed the reviewers' criticism in their detailed rebuttal and in the manuscript. I agree with the other reviewers that this work does not report an absolute first observation nor present a definitive "smoking gun" for topological superconductivity. Nevertheless, it does represent a good step forward owing to the excellent momentum and energy resolution and the careful temperature dependence. The analysis appears to be sound. I believe that the paper will be of interest to the readers of *Nature Communications*.

Authors' reply:

We thank the reviewer's comment. We appreciate the reviewer's approval that our work "represent a good step" and "analysis appears to be sound". We thank the reviewer for his/her recommendation of publication of the current manuscript in *Nature Communications*.